# Approaching Suffering in Young University Students, New Challenge for a Compassionate University: A Qualitative Study of Undergraduate Nursing Students

**DOI:** 10.3390/healthcare12242571

**Published:** 2024-12-20

**Authors:** Sonia García-Navarro, E. Begoña García-Navarro, Miriam Araujo-Hernández, Ángela Ortega-Galán, María José Cáceres-Titos

**Affiliations:** 1Department of Nursing, University of Huelva, 21071 Huelva, Spain; bego.garcia@denf.uhu.es (E.B.G.-N.); angela.ortega@denf.uhu.es (Á.O.-G.); mariajose.caceres@denf.uhu.es (M.J.C.-T.); 2District Huelva Costa, Clinical Nursing, 21007 Huelva, Spain; 3Centro de Investigación en Pensamiento Contemporáneo e Innovación para el Desarrollo Social (COIDESO), University of Huelva, 21007 Huelva, Spain

**Keywords:** support, compassion, suffering, university students, nursing students

## Abstract

Background/Objectives: Student distress is diverse and manifests itself in a variety of ways. Driven by the constant pressure to meet academic and personal expectations, many students experience a deep sense of insufficiency and hopelessness. Anxiety and depression are widespread and are often accompanied by self-critical thoughts and feelings of worthlessness. Seeking temporary relief that often exacerbates their long-term struggles, some students resort to self-harm as a way to manage their emotional pain. Methods: This study employs an exploratory qualitative design with a phenomenological approach to deeply examine students’ experiences of suffering. Using semi-structured interviews, the study captures detailed narratives from 17 purposely selected students, providing a rich understanding of their lived experiences. Adhering to COREQ guidelines (Consolidated Criteria for Reporting Qualitative Studies). Results: The analysis identified four main themes: the concept of suffering, its causes, students’ responses, and coping strategies. Eleven sub-themes were identified, highlighting the significant challenges students face, including low self-esteem, anxiety, depression, and self-harm. These problems are increased by academic stress and social pressure derived from the intensive use of social networks. The constant social comparison and the search for external validation on social networks generate feelings of inadequacy and emotional exhaustion. Students often lack effective strategies to manage these pressures. In addition, bullying and the absence of meaningful relationships in the academic environment increase their suffering. Conclusions: This study provides a novel contribution by focusing on the lived experiences of nursing students and emphasizing the importance of the compassionate university model as a framework for addressing student suffering. It highlights the urgent need to implement strategies for support and compassion in educational institutions centred on emotional management, presence, listening, and specific mindfulness and stress management programmes. Additionally, fostering healthier and more mindful use of social media is crucial to help students manage their expectations, reduce their suffering, and cultivate self-compassion.

## 1. Introduction

Suffering is a difficult phenomenon to address due to its subjective and heterogenous nature. Indeed, the term is often used as a synonym of pain; however, the two concepts are not one and the same. Pain refers to an unpleasant physical or sensorial experience [1], whilst suffering is a much broader experience that includes pain but, also, can include other types of negative experiences such as loss, lack of hope for the future, and injustice [2]. Diverse definitions have been used to try to explain suffering. Some consist of simple words, such as “extreme anguish”, “soul pain”, and “not being whole” [3], whilst other definitions comprise more complex ideas such as “a state of profound concern for situations that endanger individual wellbeing” [4] and “a state of intense awareness and emotional response towards the loss, limitation or alteration of key aspects of personal development” [5].

Suffering can be understood as an adverse and distressful experience that profoundly affects an individual as much at a psychological and physical level as at an existential level. Whilst it is not rooted in any biological cause of note, suffering is manifested through perceptible physical symptoms that affect emotions and general wellbeing [6]. Pain is not alone as a cause of suffering. Issues such as poverty, existential crisis, anxiety, stress, and social exclusion can all lead to suffering [7]. Suffering may even cause a disconnect from the aforementioned concerns, leading to a state of liminality—in other words, a state of uncertainty in which one is neither here nor there, whether physically or mentally [8]. Suffering is a personalised experience that is shaped by individual experiences of it [1] in the sense that the same situation can be tolerated well by one individual whilst being deeply distressful for another. This is seen in the way that the loss of a loved one can be lived very differently by different individuals belonging to the same family, as a function of their bond, relationship, prior experience with grief, and the coping mechanisms at hand [9]. Further, according to theories based on values, feelings, and goals, the way in which individuals suffer varies as a function of life stage [10].

Existential suffering is defined as a painful mental state that drags with it a cloud of negative emotions such as fear, loneliness, sorrow, remorse, bitterness, and shame. It is characterised by profound change and loss of control over life events [11]. The definition of spiritual suffering is more complex. It describes a clash between the beliefs, values, and hopes held by an individual and the reality lived by that individual [12] or an alteration to what could be considered the core or essence of an individual [13]. Spiritual suffering refers to the experience of being dislocated from all meaning, purpose, and connection with oneself, the divine, and others. Experiences of distress in response to uncertainty can also be considered to form a part of this type of suffering, alongside intense fear about the future. In addition, spiritual suffering is related to the struggle to find meaning in life, as well as feelings of abandonment by the divine or loss of faith [14]. Spiritual suffering and psychological disorders are intrinsically connected and reflect the way in which internal crises can profoundly impact individual mental health. This connection is manifested through mental disturbances such as anxiety, depression, stress, and behavioural eating disorders (BEDs). BEDs are mental pathologies that are characterised by nutritional imbalance and abnormal behaviour at the time of eating or during the process of weight loss [15]. Stress has been found to be tightly linked to anxiety and depression as an adaptive response when faced with a threat that may compromise wellbeing.

Starting university entails drastic life changes for those embarking on this stage. Young people go through a developmental process that is characterised by meaningful physical, emotional, and social change, which makes them particularly susceptible to various types of suffering, including social pressure, stress, and mental health issues. This vulnerability is broadened in a hedonist society that ignores these challenges and prioritises immediate pleasure and personal satisfaction.

Resilience is a trait or personal skill that facilitates coping, adaptation, and personal acceptance [16]. Resilience is promoted by active coping strategies, which help individuals to better handle stressful circumstances and develop more positive perceptions in order to overcome adversity [17]. A systematic review developed at the University of Ulster, Ireland [18], confirmed the importance of resilience in nursing students. Resilience significantly influenced stress and psychosocial morbidity, so they recommend educational strategies in the university itself that promote and improve resilience and coping strategies [19]. The university community should foster values that promote solidarity, commitment, and compassion through the implementation of activities that highlight the importance of providing support to people facing difficult situations, this value being the foundation of care. In order to inspire students to take care of themselves and generate a commitment to the relief of suffering, the project of the university as a compassionate community is generated [20]. The university, as a compassionate collective, aims to bring about a shift towards a community model that allows the integration of health and social systems, incorporating compassionate communities into health strategies and programmes as a public health approach and as a support system [21].

At the present time, modern hedonistic society is not capable of adequately tackling stressful situations and the suffering they bring with them. This incapacity is especially reflected in young people who find themselves in the middle of a cultural and social dilemma. Growing up in an environment that tends to value superficial success and avoids discomfort, young people may be particularly ill-equipped to handle adversity in their life. When these young people face inevitable challenges, such as personal failure, loss, or academic pressure, their lack of preparation and of adequate tools to cope with suffering becomes evident. Current society, by focusing on avoiding pain instead of teaching how to tackle it, leaves young people vulnerable and, oftentimes, alone in the struggle. For this reason, the incorporation of the values of support and compassion in a university model is necessary—as is working with students on values such as commitment, emotional wellbeing, and self-care [21,22].

Other authors report that young university students need to maintain good relationships with others, appreciate their environment, accept themselves and the past, set life goals, strive to develop themselves, and make their own decisions in order to benefit from good mental health. Whilst academic achievements are important for university students, in the absence of a positive attitude, students may experience suffering if they do not achieve their academic goals. Academic achievements, such as passing exams or tests, good time management, and studying many different subjects at the same time, have been found to be sources of suffering for students [23]. In addition to the very factors associated with the university context, students have described dealing with suffering caused by difficulties when adapting to the new environment. These difficulties have been detailed as living away from home, financial responsibilities, building friendships, and learning to adapt to the new university system [24].

A study carried out by an Australian university [25] examined the role of personality and contextual factors in relation to psychological wellbeing and distress amongst students throughout the semester. This study examined whether skills that help develop resilience, such as positive dialogue with oneself, self-management awareness, and meditation, can help to reduce stress and improve psychological wellbeing. Outcomes revealed that qualities such as emotional resilience and recovery capacity were important predictors of psychological wellbeing and distress. Together with another similar study [26], this study suggests that compassion for others and for oneself may enable university students to tackle, overcome, and recover from adversity and generate greater feelings of prosperity and meaning in life.

Although studies have been conducted that focused on compassion fatigue in nurses who provide support to patients living with disease [27], none have focused on the suffering of these professionals themselves. This is problematic, as it is fundamental in order to be able to promote effective nurse–patient relationships in addition to a healthy relationship with oneself. It is crucial to address suffering in this collective, given that these professionals exercise their profession whilst subjected to multiple work stressors such as being in direct contact with patient suffering and death, high workloads, staff shortages, etc. Along these lines, when examining this phenomenon, it could be useful to shine a spotlight on the earliest stage of this profession, denominated the pre-professional stage, and examine nursing degree students. With the aim of addressing the gaps discussed above, the present article proposes to, on the one hand, identify the reasons behind the suffering of nursing students and, on the other hand, uncover nursing students’ perceptions and knowledge regarding personal suffering and the suffering of others. The article also aims to identify the resources and needs of these students in order for them to deal with this experience.

## 2. Materials and Methods

### 2.1. Design

An exploratory qualitative study was designed with a phenomenological approach which examined in-depth experiences of perceived suffering in students and the meaning attributed by informants themselves to this process, as well as the approach used by students to deal with this suffering. Semi-structured interviews were conducted in order to provide detailed information on the way in which participants live such experiences [28]. The research follows the COREQ (Consolidated Criteria for Reporting Qualitative Studies) [29] guidelines for qualitative research to ensure transparency and rigor throughout the process. Key aspects addressed include study design and data analysis. Potential biases were documented, semi-structured interviews provided flexibility and consistency, and the data were transcribed and coded with inter-coder agreement. Additionally, preliminary findings were validated with participants, strengthening the reliability and validity of the results.

### 2.2. Participants

Study participants were university nursing degree students enrolled at the University of Huelva, Spain. Sampling was performed according to convenience and taking into consideration the distribution of academic years pertaining to the degree of interest. This approach was selected based on the validity and reliability of the information that could be provided by the population of interest [30] and on evidence from other similar studies [31,32]. Initially, the recruitment of informants was carried out through class delegates, avoiding the presence of any teacher, and thus preventing selection bias. Students who voluntarily wished to participate in the study had to meet the inclusion criteria of being students of adult age and had to have taken practicum courses during their undergraduate training (in order to know whether the encounter with the suffering of the patients could be influencing the mental health of the students). The final number of students was established according to discourse saturation criteria.

Participants were reached through teaching staff working within the Department of Nursing involved in the joint topics of providing support for those suffering and expressing compassion. Participating teaching staff met on a number of occasions with the research team in order to understand the research aim and, thus, proceeded to collect participant information and participate in recruitment and signed consent processes. The study adhered to international ethical principles laid out in the Declaration of Helsinki. All gathered personal information was stored in compliance with legal requirements for the protection of personal data and guarantee of digital rights (Organic Law 15/1999 of the 13 December 1999 and Organic Law 3/2018 of the 5 December 2018).

### 2.3. Data Collection

The study was performed during the first semester of the 2023–2024 academic year. Participants were requested to conduct meetings in neutral university settings not used for teaching purposes. This choice, accepted by both participants and the recruiter, creates a comfortable and safe environment that encourages open sharing of personal experiences. Initially, participating students were contacted telephonically by a member of the research team who was not involved in their recruitment. Following verification that they met inclusion criteria, participants were scheduled for interview, which was directed by a different researcher to that involved in recruitment. The different techniques employed were recorded by another team member who was experienced in the development of qualitative research techniques and was also the person responsible for taking field notes. Interviews lasted for no longer than 80 min and always started with the same leading question: “What is suffering to you?” In addition, a semi-structured interview guide was elaborated.

With a view to ensuring validity and reliability, the entire process of coding and analysing the discourse was performed independently by three members of the research team. Discrepancies were discussed until consensus was reached.

### 2.4. Data Analysis

Documents were prepared, and a hermeneutic unit was created using ©2023 Atlas.Ti Scientific Software version 23 Development GmbH (Berlin, Germany), which included all of the transcribed documents used in analysis. Interviews were analysed using the method conceived by Giorgi [33], which consists of three stages. The first stage consisted of an in-depth reading of discourse which was textually transcribed by two researchers. In the second stage, a further reading of discourse was carried out in order to extract all units of meaning. These units of meaning were labelled and grouped according to codes in line with their shared characteristics. Next, codes were reviewed and compared as a function of their shared characteristics. Following this, researchers discussed codes in order to reach a consensus regarding the final grouping of codes into categories. In the third stage, after an interpretive process was applied in group meetings, categories were grouped into general themes as a function of their shared characteristics. Finally, interpretations were made of the content within each category as a function of the phenomenon or described experience.

The present study was conducted with the help of ATLAS.Ti 23, a CAQDAS (computer-assisted qualitative data analysis software) designed to help researchers save time, perform complex processes, and provide more flexibility for the review of analytical processes. With the aim of guaranteeing the scientific rigour of the study, triangulation was performed between the different participants and researchers, reflective capacity of researchers, and pertinence of findings [34].

## 3. Results

The overall study sample comprised 17 individuals, of whom 64.7% were female and 35.3% were male. The age of participants ranged between 19 and 34, with the average age being 22.4 years (SD = ±3.4 years). With regards to previous pathologies, 58.8% of participants did not report any pathology compared with 41.2% who did. The majority of participants (76.5%) had experienced the loss of someone close to them, whilst 23.5% had not yet experienced this. Sociodemographic data pertaining to participants is presented in Table 1.

As an outcome of content analysis, Table 2 presents the main themes, sub-themes and units of meaning that emerged from the discourse. Four main themes and eleven sub-themes emerged in relation with the impact of suffering in nursing students.

### 3.1. Definition of Suffering

Informants define suffering as a negative experience that causes generalised malaise and entails different dimensions pertaining to human beings, above all, physical, emotional, and spiritual dimensions.


*“Suffering could be defined in a general way as pain, but I think that this word is routed more towards pain at an emotional and spiritual level. It is the pain of losing something or someone you love or even losing yourself, although comments made by other people from our environment can also cause a lot of pain at an emotional level.”*
(E17).

### 3.2. Reasons for Suffering

The main reasons behind the suffering of nursing students cover a wide array of social, personal, and academic situations that unchain different negative emotions and challenges.

Of these social situations, the majority pertain to the loss of relatives or pets. Further, respondents do not, only, mention death but, also, cite unresolved grief, especially in cases in which it was not possible to say goodbye, as occurred during the COVID-19 pandemic.

Interpersonal relationships also represent a substantial source of suffering to be highlighted in gathered discourse. The breakdown of friendships or romantic relationships, alongside the challenge of fitting into new social groups or communities and social pressure emerge as other reasons identified through content analysis. This leads to the emergence of unwanted solitude.


*“Young people are characterised by ‘what is expected of young people’: you have to go out partying, drinking alcohol, etc. and I don’t consider myself like that. I am a calmer person who much prefers to go out in the afternoon for a picnic, a little walk, etc. so I also feel a lot of pressure in this sense”.*
(E1).

Concern for family wellbeing that is rooted in parents divorcing or loved ones suffering from illness represents another notable reason for suffering. Respondents also mention financial challenges that impede them from living independently without depending on their parents.

At a personal level, previous personal pathologies stand out, alongside low self-esteem, combined with excessive self-demands for personal and academic success.

Finally, suffering related to the university setting itself emerges. Here, respondents mention the pressure to obtain good outcomes and attendance-related practices. These are inextricably linked to nursing degrees, with students feeling overwhelmed by their workload and that they do not have enough time. The relationship with placement tutors and the way in which the profession is exercised also stood out as motives, leading to generalised feelings of frustration and suffering in the face of insufficient attention from the course tutor, whom students feel that they cannot confront or openly discuss the issue with.


*“There are nurses who go into a room and don’t say anything, not good morning, nor how are you. They come in, give the medication and they leave. And of course, whilst I don’t agree with this, I can’t say anything because I don’t have that trust and they might feel judged, she is the one who grades me, so I keep quiet and I bite my tongue.”*
(E7).

The relationship between categories and codes is presented in Figure 1. Social situations are highlighted in yellow, personal situations in blue, and academic situations in pink.

### 3.3. Response to Suffering

Response to university suffering was expressed in informant discourse through diverse emotional and behavioural manifestations which reflect the negative impact exerted by suffering in the life of university students. Amongst the most common emotional manifestations, respondents mentioned crying and paralysis or shock, which diminished participating students’ capacity to face the situation and make decisions.

Guilt and frustration were also common emotions, with the sense of responsibility weighing heavily on respondents. Anger and impotence when faced with obstacles or situations perceived to be unfair also emerged.

With regards to emotional disorders, common responses to suffering amongst university students included stress, anxiety, and depression. All of the aforementioned are responses related to mental health and were manifested through somatisations such as headache, backache, chest or abdominal pain, intestinal problems, nausea, lack of appetites, and insomnia.

In addition, some students mentioned self-destructive behaviours as a means of dealing with suffering. Such behaviours included self-harm and alcohol and drug abuse.

### 3.4. Management of Suffering

The management of suffering in the university context implies the use of different resources and tools to manage emotional malaise and promote general wellbeing in students. Informants expressed the need to adoppt a compassionate outlook towards themselves and others. This entailed attending to perceived needs and showing concern for the wellbeing of others in addition to seeking out professional help when needed. Cultivating support networks, providing mutual emotional support, and offering physical contact, such as lending a hand or arm, when necessary, were also elements that stood out from respondent testimony.

Some participants mentioned contact with nature as a resource for reducing stress and promoting relaxation. At the same time, these participants described writing as a means of emotional expression. Active listening, meditation, and conscious breathing practices were also considered to be useful tools for managing personal suffering. This being said, these tools were not mentioned as resources for managing the suffering of others.

Despite counting on elements to manage suffering, diverse difficulties or barriers were also identified from participant discourse. Such barriers included lack of spaces set up for self-care and emotional expression, absence of psychological support, and negative patient attitudes towards being cared for, as seen through examples of refusal or disinterest. Compassion fatigue, excessive workload, ethnocentric care provision, and lack of communication were also elements identified as hurdles to effective management of suffering, with the impact of the latter being intensified in cases in which a language barrier was also present.


*“Another time there was a barrier is when I was met with a language barrier between the patient and the nurse. One of my colleagues tried to help her with Google Translate, which is a super simple tool that is free and available to everyone and I heard the nurse tell her, ‘don’t do that, it’s a waste of time’. And afterwards to see that, this same nurse, the only thing she did to get over that barrier was to talk to the person more loudly and slowly. Really? I just don’t get it, as much as you speak to them loudly and slowly, they are not going to understand if they speak another language. And it’s not just that she doesn’t do it but, also, that they judge a colleague for doing it.”*
(E11).


*“With regards to others, I think it mainly happens when I tried to ‘put myself in their shoes’ and I end up assessing their situation from my own point of view (because you can’t really put yourself in somebody else’s position) and I found it really difficult to see the issue and show compassion because, for me, it wasn’t a problem.”*
(E9).


*“The other situation in which I find it difficult to show compassion is in situations in which I considered that the other person has done something really bad, really cruel, so much so that they do not deserve compassion. Although I try to think that these criminals are not monsters, they are people and they deserve to be treated as such. They deserve to be punished and they will be, but as soon as you dehumanise them and turn them into monsters in your head, something inhuman, it is you who ends up losing, because it blinds you. You are dealing with people, whatever they have done. And this pain caused by their actions must never cloud their humanity.”*
(E9).


*“I was so overwhelmed and I was also going through a tough time, then somebody who wasn’t going through a good time either, I tried to listen to them, I tried to help them, but I couldn’t give all of me or be at the top of my game, as I wasn’t in a good place either, I felt bad, honestly, it was a bad experience because I didn’t feel good with myself, I apologised to that person, but even so, afterwards I learned that you can’t always be up to the task.”*
(E12).

Finally, a number of benefits to managing suffering were identified, such as improved quality of care, personal growth, and greater satisfaction derived from striving to alleviate another’s suffering.


*“When we don’t find a solution, we are a society with weaknesses. Feeling that another person is suffering and showing it does not form part of the system. They consider us to be robots and we can’t be robots. We feel the suffering of others, but we can’t be indifferent to it. Society turns its back on suffering. Suffering is seen as a weakness, you don’t connect with the other person. Thanks to my training, I have realised that emotions can be heightened, can be worked on, can be used to strengthen that connection. I believe in that and that I must work on myself to be able to care for others. Even if it is hard work, I believe that it is necessary to take emotions into account in order to be able to provide quality care.”*
(E16).


*“A patient died the other day and it was me who told the family. Well, I told them that the electro had come out flat. I didn’t say to them that they had died because I don’t feel that I have the skills to give this news to anybody, but I did say that the electro was flat and that means that there is not heartbeat and it’s true that I wasn’t, or that I couldn’t be compassionate with that family and stop to talk with them or pay my respects. That is all I said, I took the electro and I left. And it’s true that afterwards I thought about it and I felt really, really bad because for that family it was a moment in which they were having a really bad time and if I could have at least been there, I mean it also wouldn’t have really changed much, because they comforted each other, but a little bit more compassion would have been better, I could have listened for a little while, I don’t know.”*
(E11).


*“Compassion is important as much in normal life as in my future life as a nurse. In fact, I think it is essential. In the present, materialistic and utilitarian world, where there is no time to stop and look around and listen to others, where compassion is considered a luxury to be shown towards a select few, a waste of time, it is essential that the few compassionate people left do not lose this ability but exploit it, cultivate it, and teach it to others, because compassion is a cure, a light for those who live in the dark, and a hand to guide those in need.”*
(E3).

Figure 2 presents a relationship map that visually illustrates the connections between the principal themes, sub-themes, and units of meanings identified in the analysis. This diagram provides a comprehensive overview of how the central themes of the study are interrelated, offering a clear framework for understanding the key findings. By mapping these associations, the figure highlights the dynamic interplay between the various dimensions of student distress, shedding light on the factors that contribute to and exacerbate their experiences. This visual representation aims to enhance the reader’s comprehension of the study’s analytical depth and thematic structure.

## 4. Discussion

The suffering experienced by a person is constructed according to multiple dimensions pertaining to the body, temperament, emotional education, coping style, background, culture, family, previous experiences, and social networks together with a dimension that pertains to the internal life elements that make individuals unique and include one’s dreams, values, beliefs, hopes, memories, and fears. Further, suffering is related to the severity of affliction [35]; however, in this case, severity is determined through assessment of the importance of the problem or the way in which it stops one from being themselves. This standpoint coincides with that proposed by informants in the present study and by other previous research [36], which reveals that the main reasons underlying suffering in the lives of university students pertain to the social, personal, and academic situations they face, depicting complex personal lived reality according to the level of threat they perceive to themselves.

Our sample of informants is mostly female, given that the study population, undergraduate nursing students, is a feminised population. However, like other studies [37,38,39], our study did not reveal a significant difference in the frequency with which genders experience distress. A study developed by Peking University [39] concludes that anxiety is significantly associated with introversion and previous mental health pathologies. Anxiety levels of freshmen and sophomores were also associated with body image, alcohol consumption habits, and academic performance.

This self-perception of threat generates distress that our informants have described as causing anxiety and suffering; this coincides with a study elaborated by Australian researchers [40] where it is concluded that psychological wellbeing had strong positive and negative associations, respectively, with resilience in the workplace—in this case, the university. 

The university stage entails academic situations characterised by high levels of pressure and suffering, particularly in students for whom the cornerstone of their training is dedicated towards the care of others. Both male and female students experience high levels of stress, anxiety, depression, and emotional burnout due to the academic burden and clinical responsibilities inherent to their field [40]. Burnout can lead to a reduction in empathy and personal satisfaction [41] and negatively affects academic performance, leading to lower grades and academic dissatisfaction [42].

University students identify aspects related to the loss of loved ones and grief as having particular resonance. Different studies confirm the existence of a close relationship of life satisfaction in young people with critical life events and the attention provided to them, and with the coping mechanisms they activate [43,44]. The inexistence of coping mechanisms is associated with risk-taking behaviours such as suicide, self-harm, and serious mental problems [35]. Such aspects were also mentioned by informants in the present study as responses to suffering.

Such suffering erupts abruptly in students’ lives through stress, anxiety, self-destructive behaviours, sleep disorders, etc. This finding is reported in multiple research studies that warn about the danger of such experiences in university students and the need that they be identified and attended to [45,46,47]. Feelings of abandonment, vulnerability, and isolation are common amongst university students in many places around the world. This calls for reflection about a truly concerning reality in much need of attention [48,49,50]. Our study, like the authors described above, highlights the need to incorporate strategies that protect students from suffering by promoting their resilience and support; some have described the need to be heard and understood, and others need to connect with themselves through finding meaning in life. This re-validates the compassionate university project that contemplates different strategies, including the incorporation of mindfulness as a transversal competence [43]. The resources and abilities possessed by the sufferer, together with the resources on offer in the social setting, are essential for ensuring that suffering has as little impact as possible on the life of young people and their passage through university. Social support serves to reduce emotional distress [51], and programmes that target resilience and provide psychological support have been demonstrated to be beneficial [52] in identifying the need to favour increased resilience amongst university students studying health sciences. This demonstrates the need for spaces that favour empathy, active listening, support, and compassion towards oneself (self-compassion) and towards others.

Compassion is a crucial element when it comes to reducing suffering, given that it enables individuals to connect emotionally and provide support to those who need it. Compassion can reduce levels of anxiety and depression, subsequently improving general wellbeing [27]. Nonetheless, students often face challenges when it comes to demonstrating compassion, especially towards those who have committed mistakes or perpetrated negative acts. This hurdle is intensified in cases in which students themselves are also going through a period of substantial emotional malaise, as this limits their ability to empathise and be compassionate as much with themselves as with others [26]. Our study shows the difficulty that young university students have in alleviating their own suffering (self-compassion) with respect to their capacity to support others (compassion). In addition, lack of compassion has been associated with higher levels of stress and lower satisfaction with life amongst university students [53]. Promoting an environment that encourages self-compassion and compassion towards others may be fundamental for improving emotional health and reducing suffering in the university community [54].

The present study detected relevant patterns pertaining to the causes of suffering in students, identifying a need for interventions that target these dimensions and promote environmental support as a means of improving wellbeing and academic performance in students. Once again, the compassionate university project is validated as a community protective factor in the suffering of students. Nonetheless, some limitations should be acknowledged, including the focus of the study on a single university context, which could limit the generalisation of outcomes. Future research should explore the phenomena examined in the present study in different contexts in order to validate and expand findings.

## 5. Conclusions

This study shows that university students experience suffering due to social, academic, and personal reasons and that it accompanies them throughout their lives. A synergy between basic and university education would enable young people to manage the suffering inherent in life.

More specifically, nursing students add to this suffering the daily work with patients and their families and the suffering inherent in illness. Despite this, they lack the necessary tools to identify and adequately manage this suffering.

The present findings reveal the need to establish spaces for self-care in the university environment to support emotional expression and reduce self-destructive behaviours as a form of expression of suffering in nursing students.

The vulnerability of university students to suffering requires the academic environment to implement proposals that aim to provide support and, therefore, make suffering visible, while providing safe spaces, psychological support, and programmes that promote resilience and cultivate compassion as basic tools.

To prevent compassion fatigue, nursing students, in particular, need training that equips them with the personal resources to manage difficult situations and teaches them the skills necessary for effective support in their own suffering and in the suffering of others.

Specific training in providing support to those who are suffering, as well as places in academic and professional environments staffed by people qualified in dealing with suffering, would be essential to ensure quality care.

The construction of compassionate universities is a major step towards this goal, and institutional support and synergies with all social strata are essential to build a society that supports care in the face of suffering.

## Figures and Tables

**Figure 1 healthcare-12-02571-f001:**
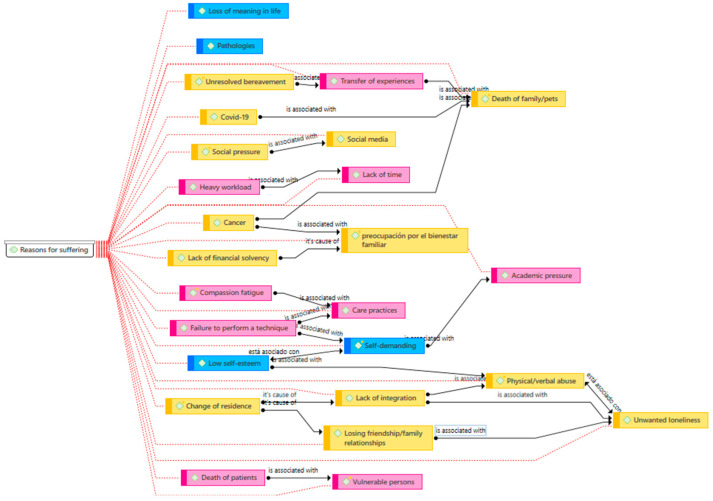
Network map of the reasons behind suffering.

**Figure 2 healthcare-12-02571-f002:**
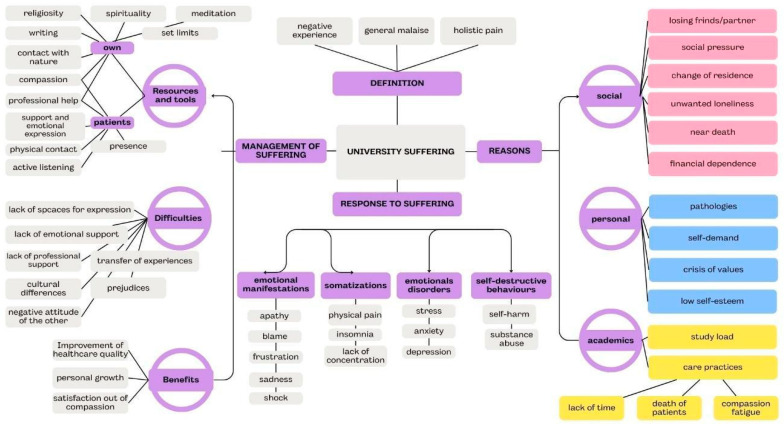
Relationship map of associations between lines of arguments, categories, and subcategories.

**Table 1 healthcare-12-02571-t001:** Participant sociodemographic data.

Participant	Sex	Age	Academic Year	Prior Pathology	Loss of a Loved One
E1	Male	23	4th	No	Yes
E2	Female	20	3rd	No	Yes
E3	Female	21	4th	Yes	Yes
E4	Female	25	4th	No	Yes
E5	Male	34	4th	No	Yes
E6	Female	21	4th	No	Yes
E7	Female	21	4th	No	No
E8	Female	22	2nd	No	No
E9	Female	26	2nd	Yes	Yes
E10	Female	27	3rd	No	Yes
E11	Female	23	4th	No	Yes
E12	Female	22	2nd	Yes	Yes
E13	Female	19	2nd	No	Yes
E14	Female	20	4th	No	Yes
E15	Female	25	4th	No	Yes
E16	Female	20	3rd	Yes	Yes
E17	Female	22	3rd	No	Yes

**Table 2 healthcare-12-02571-t002:** Themes, sub-themes, and units of meaning derived from data analysis.

Themes	Sub-Themes	Units of Meaning
Definition of suffering	Negative experience	Situation that provokes general malaise.
	Holistic pain	Manifestations at an emotional, physical, and spiritual level.
Reason for suffering	Social situations	Loss of relatives, pets, friends, or partners; unresolved grief; not being able to say goodbye; social pressure; influence of social networks; difficulty fitting in; lack of financial solvency; physical or verbal abuse; dealing with vulnerable people; concern for family wellbeing; childhood experiences associated with divorce or illness; isolation.
	Personal situations	Low self-esteem, excessive responsibility.
	Academic situations	Pressure to achieve academic outcomes, situations associated with attendance-related practices such as failure to perform a technique, lack of time, workload, compassion fatigue, and experience transference.
Response to suffering	Emotional manifestations	Isolation, crying, fear, sadness, paralysis, apathy, guilt, frustration, screaming, impotence, uncertainty, rage.
	Somatisations	Headache, backache, chest pain, stomach ache, intestinal problems, nausea, lack of appetite, insomnia.
	Self-destructive behaviours	Self-harm, substance abuse.
Management of suffering	Resources and tools	Compassionate attitude, attending to perceived needs, concern for patient wellbeing, professional support, cultivating support networks, providing emotional support, sharing experiences with colleagues/relatives, contact with nature, physical contact, writing, active listening, setting boundaries, emotional expression, meditation, not judging, self-care practices, presence, religiousness, conscious breathing.
	Difficulties	Lack of habilitated spaces, lack of psychological support, negative patient attitude (denial, superiority, other’s lack of interest, etc.), distant professional attitude, language barrier, conspiracy of silence, cultural differences, difficulty of experiencing self-compassion, professional egoism/ethnocentrism in care, compassion fatigue, lack of communication, workload, prejudices, not externalising.
	Benefits	Improved quality of attention, personal growth, satisfaction for compassion.

## Data Availability

Data will be made available upon request by the corresponding author.

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
