# Peer review of "Approaching Suffering in Young University Students, New Challenge for a Compassionate University: A Qualitative Study of Undergraduate Nursing Students"

_healthcare, 2024, doi:10.3390/healthcare12242571_

Round 1

Reviewer 1 Report

Comments and Suggestions for Authors

Dear Authors,

Thank you for the opportunity to review your manuscript. The topic is pertinent and addresses an important area of research. However, several considerations need to be addressed to enhance the clarity, coherence, and scientific rigor of the study.

Abstract

  • The abstract is well-organized but overly descriptive. Simplifying complex sentences would improve readability.
    • For example: "Students frequently lack effective coping strategies to cope with these pressures..." could be simplified to "Students often lack effective strategies to manage these pressures."
  • Emphasize the novelty of the research, such as its focus on nursing students' experiences and the compassionate university model.
  • Include specific key findings, such as the themes and sub-themes derived from the analysis, to provide a clearer snapshot of the study's contributions.

Introduction

  • Expand on the concept of a compassionate university and its relevance to the nursing field.
  • Some cited references are outdated. Integrating recent research on mental health and resilience in nursing students (post-2020) would strengthen the background.
  • The introduction could benefit from a clearer outline of the study's scope, objectives, and significance to the field of nursing and education.

Methodology

  • Provide a more detailed explanation of how participants were purposively selected, ensuring transparency about inclusion and exclusion criteria.
  • Acknowledge the potential biases of convenience sampling and suggest how these may have impacted the findings.
  • While the study adheres to COREQ guidelines, this could be discussed in more depth to clarify how rigor and reliability were ensured throughout the research.
  • Briefly elaborate on the Giorgi phenomenological method and why it was particularly suited for this study.

Results

  • Tables and figures are informative, but the thematic network map (Figure 1) could benefit from a more concise design to enhance interpretability.
  • Include a brief narrative summary of the main themes and sub-themes in the text before presenting Table 2.
  • Explore gender-based or year-of-study-based differences in experiences of suffering, as these could provide additional insights.

Discussion

  • Relate the findings more comprehensively to recent studies on student mental health and resilience in academic settings.
  • Elaborate on how promoting compassion aligns with current evidence-based strategies to improve mental health in students.
  • Address other limitations of the study, such as the homogeneity of the sample (e.g., limited to a single university and field of study).
  • Propose specific interventions, such as integrating mindfulness or peer support programs into nursing curricula.

Conclusion

  • Emphasize the practical implications of the findings, particularly for educational policymakers and university administrators.
  • Avoid repetition of detailed results; focus instead on actionable recommendations for implementing compassionate practices within the university context.

Author Response

Abstract

Comments 1: [The abstract is well-organized but overly descriptive. Simplifying complex sentences would improve readability. For example: "Students frequently lack effective coping strategies to cope with these pressures..." could be simplified to "Students often lack effective strategies to manage these pressures."]

Response 1: [Thank you for your feedback. We have revised the abstract to simplify complex sentences and improve readability. We adopted your suggestion and rephrased "Students frequently lack effective coping strategies to cope with these pressures" to "Students often lack effective strategies to manage these pressures." We appreciate your input and believe these adjustments enhance the clarity and conciseness of the text.]

Comments 2: [Emphasize the novelty of the research, such as its focus on nursing students' experiences and the compassionate university model. Include specific key findings, such as the themes and sub-themes derived from the analysis, to provide a clearer snapshot of the study's contributions.]

Response 2: [We have addressed your suggestions by emphasizing the innovative aspects of our research, particularly its focus on the lived experiences of nursing students and the implementation of the compassionate university model as a framework for addressing their suffering.]

Introduction

Comments 3: [Expand on the concept of a compassionate university and its relevance to the nursing field.

Some cited references are outdated. Integrating recent research on mental health and resilience in nursing students (post-2020) would strengthen the background.]

Response 3: [Thank you for the recommendation, we have updated the bibliography to the date given, and we have included bibliography according to the subject indicated.

Bibliography mental health and resilience in nursing students: [19,20,21]; [37,38,39,40] ]

Comments 4: [The introduction could benefit from a clearer outline of the study's scope, objectives, and significance to the field of nursing and education.]

Response 4: [Thank you very much for your advice, we have noticed that in the introduction that we had developed was not clear the concept of compassionate university and following your indications we believe that this section has gained in quality and reading clarity, on the other hand, the modifications made have allowed us to include current references of studies of nursing students who associate concepts of resilience and mental health that have enriched our justification, with all this we believe that our introduction has modified its outline as we had proposed, again thank you very much.]

Methodology

Comments 5: [Provide a more detailed explanation of how participants were purposively selected, ensuring transparency about inclusion and exclusion criteria. Acknowledge the potential biases of convenience sampling and suggest how these may have impacted the findings.]

Response 5: [Sampling in qualitative research is carried out intentionally so that the researcher is in an optimal position to collect the relevant information to answer the research question posed (Relinque et al., 2013). We have incorporated in this section the recruitment procedure, the possible biases, as well as the criteria for inclusion in a more justified way]

Comments 6: [While the study adheres to COREQ guidelines, this could be discussed in more depth to clarify how rigor and reliability were ensured throughout the research.]

Response 6: [Thank you for the comment. We appreciate the suggestion and have added a more detailed explanation of how rigor and reliability were ensured in line with COREQ guidelines. Specifically, we have elaborated on aspects such as participant selection, data collection methods, and data analysis processes. For example, semi-structured interviews were used to ensure consistency while allowing flexibility, and interviews were audio-recorded and transcribed verbatim. Coding was conducted by multiple researchers to achieve inter-coder agreement, and participant validation was carried out by sharing preliminary findings with a subset of participants to confirm accuracy. These measures underline the study's commitment to methodological rigor and the reliability of its findings.]

Comments 7: [Briefly elaborate on the Giorgi phenomenological method and why it was particularly suited for this study.]

Response 7: [The phenomenological method allows us to understand the meaning of the lived experiences of people about a particular phenomenon to be investigated, its interest is to return to the very essence of things, that is to say, to relive the experience in order to construct or reconstruct knowledge based on it, but why apply the phenomenology of Amedeo Giorgi as methodological support? applying Amedeo Giorgi's phenomenology as methodological support in the study of the experiences of suffering of young university students allowed us to understand the singular perspective from which adolescents see the real world in which they live, which is contextualized in a university reality where the typifications of the daily life of the actors will clarify the interpretation they give to their motives and actions.]

Results

Comments 8: [Tables and figures are informative, but the thematic network map (Figure 1) could benefit from a more concise design to enhance interpretability.]

Response 8: [Thank you for your comment regarding Figure 1. While we understand the need to enhance interpretability, we believe that the complexity of the relationships among the codes extracted from the informants' discourse is essential for the analysis. To facilitate understanding, we have included an explanatory paragraph before the figure that details its content and purpose. We believe this will help readers better interpret the information presented.]

Comments 9: [Include a brief narrative summary of the main themes and sub-themes in the text before presenting Table 2.]

Response 9: [In response to the request, we have included a narrative summary that outlines the main themes and sub-themes identified in the text prior to presenting Table 2. This summary highlights the complex nature of suffering among nursing students.]

Comments 10: [Explore gender-based or year-of-study-based differences in experiences of suffering, as these could provide additional insights. ]

Response 10: [We appreciate the suggestions regarding the exploration of gender differences. While we conducted a preliminary analysis of these differences using the code-document tool in Atlas.ti and identified interesting patterns, we chose not to delve deeper into this aspect in the article for two main reasons. First, the study's objective focuses on discourse analysis, emphasizing how participants narrate their experiences and construct meanings, rather than on differences in code frequencies, aligning with the interpretive design of the study. Second, the sample composition, predominantly female, reflects the reality of nursing students but limits an equitable gender comparison. This imbalance could introduce biases in the conclusions, reducing the robustness of a gender-specific analysis. These considerations led us to prioritize a global analysis of the discourses, highlighting the thematic and narrative patterns most representative of the shared experiences of the participants.]

Discussion

Comments 11: ^Relate the findings more comprehensively to recent studies on student mental health and resilience in academic settings. Elaborate on how promoting compassion aligns with current evidence-based strategies to improve mental health in students.]

Response 11: [Thank you very much for this proposal, once these studies have been incorporated in the context of the discussion of the results, not only has this section been further clarified, but the concept of a compassionate University has been validated with much more strength, for which we greatly appreciate your proposal. ]

Comments 12: [Address other limitations of the study, such as the homogeneity of the sample (e.g., limited to a single university and field of study).]

Response 12: [We have incorporated this limitation to the manuscript, in this sense it is difficult to incorporate another sample at European level, since our University has achieved this year the certification of the first compassionate university in Europe. This justifies the suitability of this study in a single university, as future lines will make comparisons with other universities with similar characteristics. ]

Comments 13: [Propose specific interventions, such as integrating mindfulness or peer support programs into nursing curricula.]

Response 13: [We have incorporated this proposal into the manuscript, supported by the evidence:

Egan, Helen, et al. "Mindfulness, self-compassion, resiliency and wellbeing in higher education: a recipe to increase academic performance." Journal of Further and Higher Education 46.3 (2022): 301-311.]

Conclusion

Comments 14: [Emphasize the practical implications of the findings, particularly for educational policymakers and university administrators.

Avoid repetition of detailed results; focus instead on actionable recommendations for implementing compassionate practices within the university context.]

[We have restructured the conlcussions, emphasising more practical proposals and avoiding the elimination of repetition.]

Reviewer 2 Report

Comments and Suggestions for Authors

The manuscript addresses an important topic, exploring suffering in university students with a focus on nursing students. It combines qualitative research and phenomenological analysis, providing unique insights. However, several aspects require improvement to strengthen the manuscript's quality and relevance.

The manuscript does not clearly outline how its findings significantly advance existing literature. The discussion relies heavily on citations that describe previously known issues.

Line 11: "fuelled" -> Should be "fueled" (American English).

Line 19: "COREQ guidelines for qualitative studies" -> Provide citation here.

Objectives could include the development of actionable recommendations to address identified issues.

Sampling: Purposive sampling may lead to bias, but this is not adequately acknowledged in limitations.

Limited details on how thematic saturation was reached.

Line 159: "adheres to COREQ guidelines" -> Consider listing some specific items from these guidelines that were followed.

Ethical considerations for dealing with students discussing sensitive issues are not sufficiently elaborated.

Line 183: Clarify why a neutral location for interviews was chosen and its significance.

Figure 1: lack of in-text citation.

Figure 2: lack of in-text citation.

Recommendations for self-care spaces and resilience programs are relevant. However recommendations are broad and lack practical specificity or implementation strategies.

Table 2 (page 7) “unwanted solitude"  => solitude or isolation?

Lines 374-426:  The discussion is overly reliant on cited literature rather than the study's findings. For example, it mentions concepts like "dimensions of suffering" but does not highlight specific quotes or data points from the interviews that support these dimensions.

Line 431: "Future research should explore..." – elaborate on what methodological adjustments would be used.

The study presents valuable data but may need improvements in the depth of analysis, practical recommendations, and clarity of writing. Addressing these issues would enhance the manuscript's quality.

Author Response

REVISOR 2

Comments 1: [The manuscript addresses an important topic, exploring suffering in university students with a focus on nursing students. It combines qualitative research and phenomenological analysis, providing unique insights. However, several aspects require improvement to strengthen the manuscript's quality and relevance.

The manuscript does not clearly outline how its findings significantly advance existing literature. The discussion relies heavily on citations that describe previously known issues.]

Response 1: [Thank you very much for this comment, which although at first may seem disturbing, has allowed us to make a constructive analysis of some sections of the manuscript that needed improvement, so we have modified part of the wording of the manuscript, and clearly we have improved the presentation of the results and the discussion, thus improving the scientific and reading quality. Thank you again for your kind words. ]

Comments 2: [Line 11: "fuelled" -> Should be "fueled" (American English).]

Response 2: [Thank you for your observation. To align with American English and improve the flow, we replaced "fuelled" with "driven," ensuring consistency and clarity in the text.]

Comments 3: [Line 19: "COREQ guidelines for qualitative studies" -> Provide citation here.]

Response 3: ^Thank you for your feedback. As this is the abstract, we have not included specific references to maintain its concise nature, which is a common practice in academic writing. However, the COREQ guidelines are appropriately referenced in the main body of the document, corresponding to citation 29. This ensures proper attribution and allows readers to locate the source for further detail.]

Comments 4: [Objectives could include the development of actionable recommendations to address identified issues.]

Response 4: [Thank you for your proposal, we have incorporated the practical recommendations in the discussion and conclusions section, as we saw that it could be more attractive for the reader.]

Comments 5: [Sampling: Purposive sampling may lead to bias, but this is not adequately acknowledged in limitations.

Limited details on how thematic saturation was reached.]

Response 5: [Thank you very much for your appreciation. We have incorporated these modifications in the methodology section of the manuscript, we have detailed why this sampling is an epistemiological characteristic of phenomenological research that does not include any bias as it is characteristic of it.

The degree of thematic saturation was reached by two conceptual models - theoretical saturation and thematic inductive saturation. The following were considered for theoretical saturation: the development of conceptual codes and observation, in the collection and analysis of data, when they generated new categories/subcategories or only indicated increasing instances. For inductive thematic saturation, the use of new codes based on each interview stood out.]

Comments 6: [Line 159: "adheres to COREQ guidelines" -> Consider listing some specific items from these guidelines that were followed.]

Response 6: [Thank you for the comment. We appreciate the suggestion and have added a more detailed explanation of how rigor and reliability were ensured in line with COREQ guidelines. Specifically, we have elaborated on aspects such as participant selection, data collection methods, and data analysis processes. For example, semi-structured interviews were used to ensure consistency while allowing flexibility, and interviews were audio-recorded and transcribed verbatim. Coding was conducted by multiple researchers to achieve inter-coder agreement, and participant validation was carried out by sharing preliminary findings with a subset of participants to confirm accuracy. These measures underline the study's commitment to methodological rigor and the reliability of its findings.]

Comments 7: [Ethical considerations for dealing with students discussing sensitive issues are not sufficiently elaborated.]
Response 7: [Given the sensitivity of the topics addressed, several measures were taken to ensure the emotional safety and well-being of participants. Before the study began, a detailed information sheet was provided to each participant, explaining the objectives, procedures, and potential implications of the research. Informed consent was obtained, emphasizing the voluntary nature of participation and the right to withdraw at any time without consequences.

Discussions were conducted in a supportive and non-judgmental environment, ensuring confidentiality and anonymity throughout the process.

The study was conducted in accordance with the Declaration of Helsinki and approved by the Institutional Review Board (or Ethics Committee) of the Comité de Ética de la Investigación de la Provincia de Huelva (Suff_Acomp, 0084-N-24). For studies not involving humans or animals, the highest ethical standards were also maintained.]

Comments 8: [Line 183: Clarify why a neutral location for interviews was chosen and its significance.]

Response 8: [Thank you for your comment. It's important to clarify that the choice of a neutral location for interviews is significant for several reasons. First, a neutral environment helps create a comfortable and safe space for participants, which can facilitate more open and honest communication. This is particularly relevant in qualitative interviews, where participants may share personal and sensitive experiences.

Additionally, a neutral location minimizes distractions and interruptions that may occur in familiar or home settings, where commitments and obligations can interfere with concentration and focus during the interview. By selecting a location outside of the participant's daily life, the aim is to eliminate potential biases and ensure that the information collected is as genuine and representative of their experiences as possible.

Finally, the neutrality of the location also helps establish a more balanced dynamic between the interviewer and the interviewee, promoting a sense of equality that can foster trust. In summary, choosing a neutral location is not merely a matter of convenience; it is essential for ensuring the quality and validity of the data obtained in the research.

Taherdoost, H. (2022). How to conduct an effective interview; a guide to interview design in research study. International Journal of Academic Research in Management, 11(1), 39-51.

Gubrium, J. F., Gubrium, J. F., Marvasti, A. B., & McKinney, K. D. (2012). The SAGE Handbook of Interview Research: The Complexity of the Craft (second ed.): SAGE Publication.

Doody, O., & Noonan, M. (2013). Preparing and conducting interviews to collect data. Nurse researcher, 20(5).]

Comments 9: [Figure 1: lack of in-text citation.]

Response 9: [Thank you for noting this. The lack of an in-text citation for Figure 1 has been addressed. We have added a reference to the figure in the text at line 279 to ensure proper integration and alignment with the document.]

Comments 10: [Figure 2: lack of in-text citation.]

Response 10: [Thank you for noting this. The lack of an in-text citation for Figure 1 has been addressed. We have added a reference to the figure in the text at line 373 to ensure proper integration and alignment with the document.]

Comments 11: [Recommendations for self-care spaces and resilience programs are relevant. However recommendations are broad and lack practical specificity or implementation strategies.]

Response 11: [Thank you very much for your input. We have incorporated recommendations as protective factors of suffering in the discussion and conclusions section. ]

Comments 12: [Table 2 (page 7) “unwanted solitude"  => solitude or isolation?]

Response 12: [Thank you for your observation. We agree that the term isolation is clearer and more precise in conveying the concept. The term in Table 2 has been updated accordingly to ensure better understanding and alignment with the study's findings.]

Comments 13: [Lines 374-426:  The discussion is overly reliant on cited literature rather than the study's findings. For example, it mentions concepts like "dimensions of suffering" but does not highlight specific quotes or data points from the interviews that support these dimensions.]

Response 13: [Thank you very much for your proposal, we have not incorporated citations in the discussion section because they are already in the results section, what we have modified thanks to your appreciation has been more references to the results described by our informants to contrast them with similarities found in the scientific evidence. ]

Comments 13: [The study presents valuable data but may need improvements in the depth of analysis, practical recommendations, and clarity of writing. Addressing these issues would enhance the manuscript's quality.]

Response 13: [Thank you very much for this comment, which, although it may seem disturbing at first, has allowed us to improve our manuscript. We have modified the introduction to make it more interesting for the reader, not only by improving the wording, but also by incorporating important quotations and improving the outline. On the other hand, we have expanded the results after the analyses proposed by the reviewers.

Thank you again for your enlightening view of our study.]

Round 2

Reviewer 1 Report

Comments and Suggestions for Authors

Dear authors,

You have now resolved all the issues raised in the previous review

Author Response

Comments 1: You have now resolved all the issues raised in the previous review

Response 1: Thank you very much for your appreciation and for the work done to improve the article.

Reviewer 2 Report

Comments and Suggestions for Authors

The authors have addressed most of my comments, and I believe the manuscript has improved. However, a thorough review of the manuscript is recommended to refine some minor areas.

  • Lines 20 and 174: COREQ should be spelled out in full when first mentioned as an abbreviation.

  • Lines 187–193: The authors have provided justification for the inclusion criteria. However, it would be clearer if they could specify how many respondents were initially targeted for the survey, as well as how many were excluded based on these criteria.

  • Discussion: Although this is not a quantitative study, the discussion could be strengthened by addressing how the sociodemographic factors presented in Table 1 (e.g., sex, age, grade, prior pathology, loss of loved ones) may influence the experience of suffering.

Author Response

Comments 1: Lines 20 and 174: COREQ should be spelled out in full when first mentioned as an abbreviation.

Response 1: Thank you very much for your appreciation, it is true that incorporating the meaning of COREQ increases reading comprehension, we have added it. Consolidated Criteria for Reporting Qualitative Studies)

Comments 2: Lines 187–193: The authors have provided justification for the inclusion criteria. However, it would be clearer if they could specify how many respondents were initially targeted for the survey, as well as how many were excluded based on these criteria.

Response 2: In qualitative research, the incorporation of informants is continuous, as we analyze until we reach discourse saturation (Taylor & Bodgan, 1984). The recruitment of the students was carried out by the class delegate of the students who met the inclusion criteria (voluntariness and having carried out clinical practices), so the delegates began the recruitment in the same way, through their WhatsApp group. All those contacted said that if (voluntariness), from the second semester of the 2nd year the clinical practices begin, so the recruitment was carried out in a way focused on these criteria, no first-year student was invited (because they have not done internships. The recruitment in the 4th grade students was greater, since they are the students who have done the most practices, I hope I have answered your question, thank you very much.

Comments 3: Discussion: Although this is not a quantitative study, the discussion could be strengthened by addressing how the sociodemographic factors presented in Table 1 (e.g., sex, age, grade, prior pathology, loss of loved ones) may influence the experience of suffering.

Response 3: Thank you very much again for your comment, we believe that you have improved the discussion thanks to your advice, we have incorporated your proposal into our manuscript in paragraph 443-449.
